# *Leishmania infantum* Axenic Amastigotes Induce Human Neutrophil Extracellular Traps and Resist NET-Mediated Killing

**DOI:** 10.3390/tropicalmed8070336

**Published:** 2023-06-25

**Authors:** Thamara K. F. Oliveira, Jullyanna Oliveira-Silva, Leandra Linhares-Lacerda, Vanderlei da Silva Fraga-Junior, Claudia F. Benjamim, Anderson B. Guimaraes-Costa, Elvira M. Saraiva

**Affiliations:** 1Laboratório de Imunologia das Leishmanioses, Instituto de Microbiologia Paulo de Góes, Centro de Ciências da Saúde, Universidade Federal do Rio de Janeiro, Rio de Janeiro 21941-902, Brazil; thamarafonseca2@gmail.com (T.K.F.O.); jullyannaoliveira@yahoo.com.br (J.O.-S.); leandralacerda@gmail.com (L.L.-L.); 2Laboratório de Imunologia Molecular e Celular, Instituto de Biofísica Carlos Chagas Filho, Centro de Ciências da Saúde, Universidade Federal do Rio de Janeiro, Rio de Janeiro 21941-902, Brazil; fragajunior@biof.ufrj.br (V.d.S.F.-J.); cfbenjamim@biof.ufrj.br (C.F.B.)

**Keywords:** NETs, *Leishmania*, neutrophils, signaling pathways, ROS, parasites, innate immunity

## Abstract

Neutrophils are multifaceted cells that, upon activation, release meshes of chromatin associated with different proteins, known as neutrophil extracellular traps (NETs). *Leishmania amazonensis* promastigotes and amastigotes induce NET release, and we have identified the signaling pathways involved in NET extrusion activated by promastigotes. Amastigotes maintain the infection in vertebrate hosts, and we have shown the association of NETs with amastigotes in human biopsies of cutaneous leishmaniasis. However, the interaction of amastigotes and neutrophils remains poorly understood. Our study aimed to characterize the pathways involved in the formation of NETs induced by axenic amastigotes from *L. infantum,* the causal agent of visceral leishmaniasis. Human neutrophils pretreated with signaling pathway inhibitors were incubated with amastigotes, and NET release was quantified in the culture supernatant. Amastigote viability was checked after incubation with NETs. We found that the release of NETs by neutrophils stimulated with these amastigotes requires the participation of elastase and peptidyl arginine deaminase and the involvement of PI3K, ROS, and calcium. Moreover, amastigotes are not susceptible to NET-mediated killing. Altogether, these findings improve our comprehension of the signaling pathways implicated in the interaction between amastigotes and human neutrophils.

## 1. Introduction

Leishmaniasis comprises a group of highly concerning diseases spread worldwide, primarily in tropical and subtropical areas. Among the different *Leishmania* species that can cause disease, *Leishmania infantum* is particularly relevant due to the clinical manifestations experienced by those infected with this parasite. *L. infantum* is the causative agent of visceral leishmaniasis (VL), the most severe form of the disease and fatal if not treated [1]. Patients with VL have their clinical condition worsened due to factors that promote cell activation and the secretion of mediators with pro-inflammatory and immunomodulatory effects, which is correlated with the inability to resolve the infection [2]. The parasite is transmitted to the vertebrate host through the bite of an insect vector, which is thought essential to disease establishment [3]. After the transmission and infection of macrophages, promastigotes of *Leishmania* transform into amastigotes, the replicative form of the parasite. Much attention has been dedicated to studying the interaction between promastigotes and the vertebrate host immune system, and still less is known about the amastigotes stage.

Neutrophils are specialized cells of the innate immune system that play a critical role in fighting infections. These cells harbor a remarkable amount and variety of proteins and enzymes essential for their development, migration, and pathogen clearance. For example, neutrophils from individuals with defects in the NADPH oxidase system are inefficient in killing pathogens, and these patients are susceptible to recurrent infections by bacteria and fungi, highlighting the importance of these cells and the production of reactive oxygen species (ROS) by the NADPH oxidase [4,5]. These cells have a wide repertoire of effector functions, including the release of inflammatory mediators and neutrophil extracellular traps (NETs) [6]. NETs are meshes of chromatin associated with neutrophil proteins released upon the activation of these leukocytes. In the last few decades, it has become clear that NETs function in the containment and elimination of pathogens [6,7], but they can also be deleterious to the host if not well regulated [8,9].

Neutrophils are rapidly mobilized to the site of *Leishmania* infection, and it has been proposed to impact disease outcome. The interaction of neutrophils and *Leishmania* promastigotes has been extensively studied. It has been shown that *L. infantum* promastigotes activate the production of ROS and the release of MPO and elastase, inflammatory cytokines, and NETs [10]. Promastigotes of all tested Leishmania species, including *L. amazonensis, L. major, L. infantum, L. mexicana*, and *L. donovani*, as well amastigotes of *L. amazonensis*, are potent inducers of NETs [11,12,13,14,15]. NET release by neutrophils stimulated with promastigotes of *L. amazonensis* requires neutrophil elastase activity, reactive oxygen species (ROS) production, and the enzyme peptidyl arginine deaminase (PAD) [16]. In addition, the activation of protein kinase c (PKC), extracellular signal-regulated kinase (ERK), the isoforms of phosphoinositide 3-kinases (PI3k), PI3K-γ, and PI3K-σ, and calcium mobilization have also been implicated in NET production upon *L. amazonensis* promastigote activation [17]. NETs are present in the lesions of patients with cutaneous leishmaniasis, where amastigotes have been observed trapped within these structures [18]. Furthermore, genes associated with NET production are differentially regulated in VL patients with active disease, asymptomatic individuals, and uninfected controls [19]. It is important to emphasize that asymptomatic individuals and those with visceral leishmaniasis demonstrate heightened levels of sera cell-free DNA, which serves as a marker of NETs, suggesting that NETs play a critical role in the infection.

In this study, we sought to analyze the mechanisms behind NETs’ release upon activation with amastigotes of *L. infantum*. We demonstrate that this parasite induces the release of NETs through the activation of PI3K, ROS production, the mobilization of calcium and neutrophil elastase, and PAD activation. Moreover, amastigotes are not susceptible to NET-mediated killing.

## 2. Materials and Methods

### 2.1. Isolation of Human Neutrophils

Human neutrophils were obtained from healthy blood donors by density gradient centrifugation [11]. Neutrophils were resuspended in RPMI 1640 medium (Sigma, St. Louis, MA, USA) and used immediately. All procedures involving human blood were performed in accordance with the guidelines of the Research Ethics Committee (Hospital Universitário Clementino Fraga Filho, UFRJ, Brazil), protocol number: 4261015400005257.

### 2.2. Parasite Culture

Promastigotes of *L. infantum* (MCAN/BR/2008/1112) kindly donated by Dr. Elaine Coimbra (Universidede Federal de Juiz de Fora, MG, Brazil) were maintained in Schneider’s Insect medium (Sigma, St. Louis, MA, USA) supplemented with 10% fetal bovine serum (FBS, Cultilab; Sao Paulo, Brazil) and 1% penicillin/streptomycin (Sigma, St. Louis, MA, USA). *L. infantum* promastigotes were differentiated into axenic amastigotes in Grace’s Insect medium (Sigma, St. Louis, MA, USA) supplemented with 10% FBS and maintained at 32 °C in a shaker incubator (Nova Técnica^®^, Piracicaba, Brazil) for 5 days [20]. The parasites were washed three times with phosphate-buffered saline (PBS; Lonza, Walkersville, MD, USA) and resuspended in RPMI 1640 medium (Lonza, Walkersville, MD, USA).

### 2.3. Axenic Amastigote Characterization

The total RNA was extracted from 10 × 10^7^ *L. infantum* amastigotes or promastigotes with TRIzol reagent (Sigma, St. Louis, MA, USA) following the manufacturers’ procedures. Using the High-Capacity cDNA Reverse Transcription Kit (Life Technologies, Carlsbad, CA, USA) for cDNA synthesis, 1.5 µg of RNA was used. Quantitative real-time RT-PCR was performed with the SYBR-green fluorescence quantification system. The PCR cycling parameters were 95 °C (10 min) and then 40 cycles of 95 °C (30 s) and 60 °C (1 min), followed by the standard denaturation curve. The primer sets were as follows [21]:

α-tubulin forward (F): 5′-TCAAGTGCGGCATCAACTAC-3′;

α-tubulin reverse (R): 5′-GAGTTGGCAATCATGCACAC-3′;

Amastin like 1 (F): 5′-AGGTGTGATGTGCTGAACGACGAT-3′;

Amastin like 1 (R): 5′-ACGGGAGCATCAGGAAGATGATGT-3′;

Amastin like 2 (F): 5′-CATCTTCGTGTACGGCTTTGCGTT-3′;

Amastin like 2 (R): 5′-TTCGGTAAGTCACCACCATGAGCA-3′;

eIF-3 (F): 5′-TCAAGACGCCCTTCACCACTTTCT-3′;

eIF-3(R): 5′-AGCGGTTAATCACCTCGTTCTCGT-3′.

All targets to α-tubulin relative expression were calculated using the comparative Ct method. Accession number and product size: α-tubulin: XM_003392311.1-91 bp; Amastin like 1: XM_003872586.1-106 bp; Amastin Like 2: XM_003886537.1-146 bp; eIF-3: XM_003865473.1-189 bp.

### 2.4. NET Induction Assay

The neutrophils (10^6^) were incubated or not with different stimuli for 90 min at 35 °C and 5% CO_2_. After centrifugation at 400× *g* for 10 min, NETs were recovered in the culture supernatant and then further centrifuged at 2800× *g* for 10 min. The DNA concentration was determined using Quant-iT PicoGreen dsDNA Assay Kit (Thermo Fisher Scientific, Eugene, OR, USA) at excitation and emission wavelengths of 485/538 nm on a SpectraMax^®^ Paradigm (Molecular Devices, San Jose, CA, USA). Pharmacological inhibition was performed by pretreating the neutrophils for 30 min with the following reagents: apocynin (1 μM, Sigma, St. Louis, MA, USA), DPI (Diphenyleneiodonium, 32 μM, Sigma-Aldrich, St. Louis, MA, USA), chloroamidine (12 μM, Cayman Chemical, Ann Arbor, MI, USA), GSK-484 (15 µM, Cayman Chemical, Ann Arbor, MI, USA), elastase inhibitor III (MeOSuc-Ala-Ala-Pro-Val-CMK, 10 µM, Calbiochem, La Jolla, CA, USA), myeloperoxidase inhibitor I (300 nM, Calbiochem, La Jolla, Ca, USA), PD98059 (50 µM, Sigma, St. Louis, MA, USA), wortmannin (200 nM, Sigma, St. Louis, MA, USA), AS604850 (10 µM, Sigma, St. Louis, MA, USA), IC87114 (1 µM, Cayman Chemical, Ann Arbor, MI, USA), and BAPTA-AM (10 µM, Calbiochem, La Jolla, CA, USA).

### 2.5. Immunofluorescence

Neutrophils (5 × 10^5^) adhered to 0.001% poly-L-lysine-coated slides were stimulated with CSFE-labeled amastigotes (10^5^) for 90 min at 35 °C and 5% CO_2_, and fixed in 4% formaldehyde (Sigma, St. Louis, MA, USA). The NETs were stained with rabbit anti-human elastase antibody (1:400 dilution; Calbiochem, La Jolla, CA, USA) diluted in blocking solution (PBS, 3% BSA) and Alexa Fluor 546-goat anti-rabbit secondary antibody (1:300 dilution; Thermo Scientific, Eugene, OR, USA) for 40 min. Slides mounted in ProLong Gold Antifade Mounting media with DAPI (10 μg/mL; Thermo Fischer, Eugene, OR, USA) were imaged in a Zeiss DMi8 confocal microscope (Zeiss, Oberkochen, Germany) and analyzed using an LAS AF imaging program.

### 2.6. Amastigotes Labeling with Carboxyfluorescein Succinimidyl Ester (CFSE)

Amastigotes (10^7^) were labeled with CFSE (0.5 μM; Invitrogen, Waltham, MA, EUA) for 15 min at 35 °C and 5% CO_2_, followed by the addition of RPMI 1640 (without phenol, Lonza, Walkersville, MD, USA) containing 20% FBS and kept on ice for 15 min. Then, the parasites were washed three times and resuspended in RPMI.

### 2.7. ROS Production

Neutrophils (10^6^) were pre-incubated or not with DPI (32 μ M) or apocynin (1 µM) for 30 min at 35 °C, 5% CO_2_. Afterwards, the neutrophils were incubated with amastigotes (10^6^) or PMA (100 nM), and dihydrorhodamine 123 (DHR, 1.2 μ M, Sigma, St. Louis, MA, USA) was added. The fluorescence intensity of individual cells was analyzed by counting 50,000 events by flow cytometry (BD FACSCaliburTM Flow Cytometer—BD Biosciences, Franklin Lakes, NJ, USA). The results are expressed as % of DHR123 high neutrophils considered to be high ROS producers and as the mean fluorescence intensity (MFI).

Alternatively, neutrophils (5 × 10^5^/well) were pre-incubated or not with wortmannin (200 nM), AS604850 (10 µM), IC87114 (1 µM), or BAPTA-AM (10 µM) for 30 min at 35 °C, 5% CO_2_. Afterwards, the neutrophils were stimulated with amastigotes (2.5 × 10^6^) or PMA (100 nM) in the presence of dihydrorhodamine 123. The cells were incubated for 120 min and the reaction was read at 500/540 nm (excitation/emission) using a spectrofluorometer reader (Molecular Devices, Sunnyvale, CA, USA).

### 2.8. Production of NETs-Enriched Supernatants

Neutrophils (8 × 10^6^) were incubated with axenic amastigotes (8 × 10^5^) for 4 h at 35 °C and 5% CO_2_, and the NETs recovered in the culture supernatant after 400× *g* for 10 min were further centrifuged at 2800× *g* for 10 min. The DNA concentration was determined using a Quant-iT PicoGreen dsDNA Assay Kit (Thermo Fisher Scientific) at 485/538 nm excitation/emission on a SpectraMax^®^ Paradigm (Molecular Devices, Sunnyvale, CA, USA). A standard curve with lambda DNA (Thermo Fisher) was used to determine the concentration of DNA in the NET-enriched supernatants.

### 2.9. NETs’ Toxicity to Amastigotes

Amastigotes (10^6^) were incubated in a 96-well plate (Jet Biofil, Guangzhou, China) with different concentrations of NET-rich supernatant (50, 100, 200, or 500 ng/mL) at 35 °C, 5% CO_2_ for 4 h. Next, alamarBlue (Invitrogen, Waltham, MA, EUA) was added and cultures were incubated for 3 h at 35 °C, 5% CO_2_. The reaction product was quantified at 530/590 nm (excitation/emission) on a SpectraMax^®^ Paradigm (Molecular Devices, Sunnyvale, CA, USA). The data are presented as a percentage of viable amastigotes in relation to the untreated control. The parasite viability was also assayed by treating amastigotes (10^6^) with different concentrations of NET-rich supernatant (100, 500, and 1000 ng/mL) at 35 °C, 5% CO_2_ for 5 h. After this period, propidium iodide (PI; 10 μg/mL; Sigma, St. Louis, MA, USA) was added and 100,000 events were analyzed using flow cytometry (FACSCalibur, BD, Franklin Lakes, NJ, USA). The results were expressed as the percentage of dead amastigotes.

### 2.10. Statistical Analysis

The data were analyzed using GraphPad Prism 8.0 software (GraphPad Software Inc., La Jolla, CA, USA). The following tests were employed: (i) unpaired t test or (ii) one-way ANOVA (analysis of variance) for more than two groups, with Fisher’s LSD (least significance difference) as a post-test. The results were normalized (fold over control) in relation to the untreated and non-stimulated controls (Ctrl). Differences between the data were considered significant when *p* < 0.05.

## 3. Results

### 3.1. Axenic Amastigotes Characterization

To confirm the effective transformation of promastigotes into amastigotes, we assessed the expression of amastin genes that encode for amastigotes’ surface antigens [21]. We quantified the mRNAs for Amastin 1, Amastin 2, and eIF3 using qPCR and normalized them against the expression of α-Tubulin. Our results showed an increase in the relative expression of the Amastin 1 and Amastin 2 genes in the axenic amastigotes compared to the promastigotes (Figure 1A,B). As a control, we also tested the expression of eIF3, which is predominantly expressed in *Leishmania* promastigotes [22]. Promastigotes express more eIF3 mRNA compared to amastigotes (Figure 1C), thus confirming the in vitro differentiation of promastigotes into amastigotes.

### 3.2. Axenic Amastigotes Induce NET Release in Human Neutrophils

We then investigated whether axenic amastigotes could induce the formation of NETs. Our results show that axenic amastigotes induced the release of NETs in a proportion-dependent manner (Figure 2A). When comparing the control to the 1:1 and 5:1 ratios of amastigotes to neutrophils, there was a significant increase in the release of NET-DNA. Specifically, the amount of NET-DNA released was 2.6 times more at the 1:1 ratio and 6.7 times more at the 5:1 ratio.

To visualize the characteristic NETs’ morphology, the neutrophils were stimulated with amastigotes and then stained with a fast panoptic stain. Extracellular fiber-like structures were observed entangling the amastigotes (Figure 2B). In addition, immunofluorescence analysis revealed NETs released by neutrophils upon parasite activation (Figure 2C–F). These findings suggest that the interaction between the axenic amastigotes of *L. infantum* and human neutrophils can induce the release of NETs, subsequently trapping the parasite.

### 3.3. Reactive Oxygen Species (ROS) Contribute to the Release of NETs Induced by Amastigotes

Reactive oxygen species production is involved in the NET release induced by *Leishmania* promastigotes [16]. We investigated whether ROS would participate in the NET release by axenic amastigotes of *L. infantum*. Initially, we analyzed ROS production by neutrophils stimulated with amastigotes using flow cytometry. We found that amastigotes increased ROS production (Figure 3A,B), and the frequency of DHR123 high neutrophils was diminished by pretreatment with DPI (97%) and apocynin (77.2%). Similarly, the fluorescence intensity increased after amastigote interaction with neutrophils (Figure 3C), and the pretreatment of neutrophils with ROS inhibitors significantly reduced the fluorescence intensity after amastigote stimulation (61.6% reduction with DPI and 38.1% with apocynin). To assess the role of ROS in the release of NETs induced by amastigotes, the neutrophils were pretreated with DPI or apocynin, followed by the testing of the NETs release. Pretreatment with apocynin inhibited 44% and DPI 15.2% of the NET release induced by amastigotes (Figure 3D). As expected, PMA-induced NET was blocked by both inhibitors. These data confirm that *L. infantum* axenic amastigotes induce ROS production in human neutrophils, and amastigote-induced NET release is partially dependent on ROS.

### 3.4. Chromatin Decondensation Promoted by Neutrophil Elastase and Peptidyl Arginine Deiminases (PAD) Contribute to Amastigotes-Induced NET Release

Neutrophil elastase (NE) and myeloperoxidase (MPO) aid in chromatin decondensation for NET formation. To investigate their role in the NET release induced by amastigotes, we pretreated neutrophils with an elastase III inhibitor (ELAi) and found a significant reduction (23%) in NET release (Figure 4A). Similarly, PMA-activated neutrophils pretreated with ELAi showed a 36% reduction in NET release (Figure 4A). However, MPO inhibition did not decrease the NET release induced by amastigotes (Figure 4B). These results suggest that NE plays a role in the NET generation induced by amastigotes. Peptidyl arginine deiminases (PADs) can also aid in chromatin decondensation through histone citrullination. To determine whether PADs are involved in NET release induced by axenic amastigotes, we used the non-selective inhibitor chloramidine (Cl-A) and the specific PAD-4 inhibitor GSK-484 (GSK). Neutrophils pretreated with Cl-A and stimulated with amastigotes showed a 15.8% inhibition of NET release, and PMA-induced NETs were inhibited by 22.6% (Figure 4C). GSK resulted in a significant reduction of 29.4% in the NET release induced by amastigotes (Figure 4D). These results suggest that the release of NETs induced by axenic amastigotes depends on total PAD and PAD-4 enzymes.

### 3.5. NET Generation by Amastigotes-Stimulated Neutrophils Requires Calcium, but Not ERK Signaling

To investigate whether the induction of NETs by amastigotes is also dependent on calcium mobilization, we pretreated neutrophils with BAPTA, a calcium chelator commonly used to disrupt intracellular calcium signaling [23]. Our results demonstrate that the pretreatment of neutrophils with BAPTA significantly reduced NET induction by amastigotes (59.5%) and PMA (63.6%), indicating that calcium is essential for amastigotes to induce NETs release (Figure 5A). To further understand the signaling pathways involved in the induction of NETs by axenic amastigotes of *L. infantum*, we evaluated the role of ERK signaling using PD98, a selective inhibitor of ERK signaling. Surprisingly, our results showed that PD98 did not affect amastigote-induced NET release, indicating that ERK signaling is not involved in this process (Figure 5B). However, consistent with previous reports, the inhibition of ERK signaling led to a significant reduction (28.8%) in NET extrusion by PMA-stimulated neutrophils (Figure 5B).

### 3.6. PI3K Signaling Controls ROS Production and NET Extrusion by Amastigote-Stimulated Neutrophils

To investigate the underlying mechanisms involved in releasing NETs induced by amastigotes, we assessed the role of PI3K and its isoforms, PI3Kγ and PI3Kδ, using pharmacological inhibitors. Neutrophils were pretreated with wortmannin, a non-selective PI3K inhibitor, AS-604850, a specific inhibitor of PI3Kγ, and IC87114, a specific inhibitor of PI3Kδ, and then stimulated with amastigotes or PMA. Our results showed that the release of NETs induced by amastigotes depends on the total PI3K, as wortmannin reduced NET extrusion by 25.1% (Figure 6A). Moreover, the participation of PI3Kγ and PI3Kδ was also observed. AS-604850 reduced NET release by 29.4% (Figure 6B) and IC87114 by 15.2% (Figure 6C). PMA-induced NET extrusion was also inhibited by all three drugs (Figure 6A–C). Thus, our findings suggest that the total PI3K, PI3Kγ, and PI3Kδ pathways are involved in the mechanisms underlying NET release by human neutrophils after interaction with axenic amastigotes.

The PI3K signaling pathway controls oxidase activation and, consequently, the production of ROS in neutrophils [24]. To investigate how PI3K and its isoforms, along with calcium mobilization, affect the production of ROS, neutrophils were treated with pharmacological inhibitors, and ROS production was analyzed (Figure 6D). All PI3K inhibitors completely abrogated ROS production induced by the parasites (Figure 6D). On the other hand, there was no difference in the ROS production in BAPTA-treated neutrophils. These findings suggest that ROS production in neutrophils induced by axenic amastigotes occurs downstream of the PI3K signaling pathway.

### 3.7. NETs Are not Toxic to Axenic Amastigotes of L. infantum

Next, we evaluated whether NETs could kill amastigotes by propidium iodide labeling through flow cytometry (Figure 7A) or alamar blue assay (Figure 7B). We found that NETs had no toxic effects on amastigotes, regardless of the concentrations tested. Positive controls for both assays were conducted using parasites killed through formaldehyde fixation.

## 4. Discussion

Our group has proven that different species of *Leishmania* can trigger NET formation in human, cat, and dog neutrophils [11,15,25]. Previously, we mainly evaluated promastigotes and examined *L. amazonensis* amastigotes purified from mice cutaneous lesions [11,12]. Subsequently, several other groups have confirmed the release of NETs from human and murine neutrophils and the HL-60 cell line after *Leishmania* stimulation [10,13,14,26,27]. In this study, we extended this research by showing that *L. infantum* axenic amastigotes induce and are trapped by NETs, which exhibit the typical NET morphology and elastase associated with the DNA scaffold. Amastigotes are the intracellular forms of the parasite that maintain infection in the vertebrate host. However, extracellular amastigotes have been observed in vivo [18,28,29]. Our group has demonstrated extracellular amastigotes in the skin lesions of cutaneous leishmaniasis intermingled with NETs [18].

Axenic amastigotes are a feasible way to obtain viable parasites in a fast and controlled manner, free of host contaminants, without the use of animals, and without the need to wait for the development of lesions. Particularly for *L. infantum*, there is no methodology for amastigote purification from infected visceral organs. For these reasons, axenic amastigotes are currently being used in various studies on drug resistance, vaccine development, gene expression identification, and cell interaction assays. They have proven effective in these research areas [28,29]. Amastin is the primary and most abundant protein expressed in the amastigotes of different *Leishmania* species, including *L. infantum* [30,31]. An immunoproteomic approach identified it solely in the extract of amastigotes-like cultures, not in the promastigote stage of *L. infantum* [32]. To ensure we were working with amastigotes, we quantified the expression of amastin. As expected, our axenic amastigotes showed a significant increase in the gene expression of amastin 1 and 2 compared to the promastigotes.

The capacity of amastigotes to induce NET formation has been previously studied [11], and the potency in activating the release of NETs seems to vary among *Leishmania* species [11,13]. In this study, we observed that the amastigotes of *L. infantum* induce the release of these structures at a ratio of 1 neutrophil to 1 parasite. In contrast, the amastigotes of *L. amazonensis* can activate NET release even when 10 times fewer parasites are used, as observed in previous work [11]. Compared to controls, both the promastigotes and amastigotes associated with visceral leishmaniasis seem less potent than *L. amazonensis*, a species primarily associated with cutaneous leishmaniasis [11,13]. For example, *L. donovani* promastigotes only start inducing the formation of NETs when five parasites are used per neutrophil [13]. Differences in the assays could account for the different observations. For example, Gabriel and colleagues stimulated neutrophils for only 30 min, while Oualha and colleagues incubated them for 18 h [10,13]. It would be interesting to compare different species of *Leishmania* associated with different clinical manifestations in a more controlled and standardized protocol to conclude this matter.

Due to the potential as targets for therapy, the signaling pathways involved in producing neutrophil extracellular traps have gained attention in recent years. NET release involves intracellular signaling and nuclear events to allow for chromatin decondensation. Since the first reports in the field, the production of reactive oxygen species has been implicated in the release of NETs [33]. Chromatin decondensation can be accomplished by elastase activity, MPO, and PAD4 [34,35,36]. The involvement of PKC, ERK, PI3K, and calcium mobilization has also been shown during NET formation. Today, we know that different stimuli may trigger different pathways for NET release [37]. Our team has shown that *L. amazonensis* can induce NET extrusion through ROS-dependent or ROS-independent mechanisms. The early/rapid activation of NETs release does not require ROS production by neutrophils [16,17]. However, the late-stage production of NETs induced by *L. amazonensis* is ROS- and nitric oxide-dependent [16]. ROS dependence could be *Leishmania* species-specific, since these free radicals are not needed for NET release induced by *L. donovani* promastigotes [13]. Amastigotes of *L. infantum* induce ROS production that is entirely abolished by apocynin pretreatment. However, our results show that the blockage of NADPH oxidase production of ROS only partially inhibited NET production. Likewise, ROS are partially responsible for NET extrusion upon the activation of neutrophils with *Candida albicans* or Group B *Streptococcus* [37]. Here, we cannot exclude the possible participation of nitric oxide and mitochondrial-derived ROS. Moreover, parasites can generate reactive oxygen species that can assist neutrophils in producing NETs, as evidenced by *Candida albicans*. [37]. In fact, it has been shown that *Leishmania* parasites can produce ROS [21].

For NET release, chromatin decondensation is essential. It has been shown that elastase migrates from the granules toward the nucleus upon ROS production, followed by MPO, and both enzymes participate in chromatin decondensation [34,38,39]. We reported that elastase, not MPO, participates in the NET extrusion induced by promastigotes [16]. We observed the same pattern with amastigotes. Post-transcriptional modifications on histones may also contribute to chromatin decondensation. PAD enzymes are calcium-dependent and promote the hypercitrullination of histones, allowing for chromatin decondensation [40]. Since we observed a blockage of NETs extrusion when a calcium chelator was used, we asked whether PAD enzymes would be involved in this process. As expected, the inhibition of these enzymes decreased the NET extrusion induced by promastigotes [16] and axenic amastigotes. The participation of PAD in NET production is a matter of debate. Some authors have shown that although citrullination occurs during neutrophil activation by different stimuli, it is not crucial for NET release [37]. In our case, the inhibition of PAD4 decreases NET production, pointing to a role for this enzyme. In this study, we did not investigate the effect of other post-translational modifications in NET formation, which has also been proposed to be important [40,41,42,43]. It is possible that other epigenetic modifications, besides PAD enzyme activity, may impact neutrophils’ ability to produce NETs in response to *Leishmania*.

Our study highlights the importance of PI3K and its isoforms, γ and δ, and calcium mobilization for generating NETs by amastigotes. Blocking PI3K pathways impacts the release of NETs triggered by various factors, including *L. amazonensis* promastigotes, immune complexes, *S. aureus*, and fMLP [17,44,45,46]. Unlike what we have previously observed for promastigotes, the inhibition of both PI3Kδ and PI3Kγ completely abrogated ROS production stimulated by amastigotes, showing that these isoforms are activated upstream of NADPH oxidase. The involvement of ERK was excluded for amastigotes’ ability to induce NET release. Thus, signaling pathways differ depending on the species or parasite stage. According to reports from our previous work, the induction of NETs by *L. amazonensis* promastigotes relies on the ERK pathway [17]. In contrast, the induction of NETs by *L. infantum* amastigotes is not affected by ERK. This difference may be attributed to the absence of lipophosphoglycan (LPG) in amastigotes [47], which are present in promastigotes and involved in NET release [11]. Additionally, *L. donovani* LPG was not involved in NET release, probably due to inter and intraspecies polymorphisms of this promastigote surface glycoconjugate [13].

Calcium is a fundamental mediator of neutrophil functions, such as degranulation, cytokine release, migration, and ROS production [48,49]. Similarly, intracellular calcium plays a crucial role in the NET formation induced by PMA, IL-8, nigericin, and urate crystals [23,37,45,50,51]. We have shown that calcium chelation inhibits NET formation by a ROS-independent pathway in neutrophils stimulated by *L. amazonensis* promastigotes [17]. BAPTA-AM treatment effectively abolishes neutrophil NET release due to amastigote stimulation, leaving ROS production unaffected. Since the PAD4 enzyme requires calcium to perform its activity [52], the inhibition of calcium by BAPTA-AM treatment may prevent the citrullination of histones and, thus, chromatin decondensation.

The first function attributed to neutrophil extracellular traps was the ability to catch and kill pathogens [6]. Since then, it has been shown that NETs are able to bind and kill bacteria, fungi, and protozoa parasites, as well as to inactivate viral particles [6,11,53,54]. Here, we found that NETs are harmless to amastigotes of *L. infantum*. We have previously shown that histones associated with human neutrophil traps kill the promastigotes of *Leishmania amazonensis* [11]. However, parasites can escape NET-mediated killing. For example, 3′-nucleotidase/nuclease expressed by promastigotes allows parasites to cleave NETs and escape NET-mediated killing [12]. A sandfly salivary nuclease is egested together with the promastigotes and, through the same mechanism, disrupts the NET structure, allowing the survival of *L. major* promastigotes [55]. Additionally, lipophosphoglycan from *L. donovani* renders this parasite stage resistant to NET [13]. The mechanisms behind amastigote resistance still need to be evaluated. However, it has been shown that amastigotes are resistant to histone toxicity [56], which could explain our results.

## 5. Conclusions

In this study, we show that amastigotes of *L. infantum* trigger NET formation from human neutrophils, and we establish here the signaling pathways implicated. Amastigotes induce the production of reactive oxygen species and these molecules are involved in NET release. We demonstrate the participation of elastase, PAD, PI3K, and calcium mobilization on the production of NETs in response to the parasites. Moreover, NETs are incapable of killing the amastigotes.

## Figures and Tables

**Figure 1 tropicalmed-08-00336-f001:**
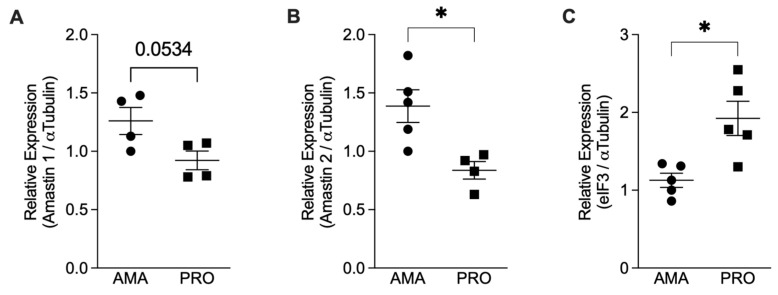
Relative mRNA expression analysis comparing *Leishmania infantum* amastigotes (AMA) and promastigotes (PRO). Total RNA from axenic promastigotes (PRO) and amastigotes (AMA) was used to prepare cDNA, and the relative expression of the genes for Amastin 1, Amastin 2, α-Tubulin, and elF3 was determined by qPCR. Results presented as mean ± SEM. (**A**) Amastin 1/α-Tubulin, (**B**) Amastin 2/α-Tubulin, and (**C**) eIF3/α-Tubulin observed in amastigotes and promastigotes of two independent experiments; * *p* < 0.05. Black circles correspond to amastigotes and black squares to promastigotes; Each symbol shows the result of a different amastigote or promastigote culture.

**Figure 2 tropicalmed-08-00336-f002:**
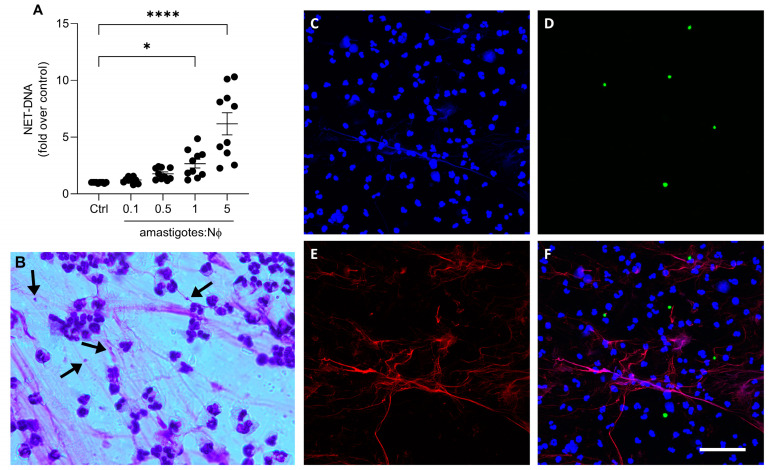
Axenic amastigotes induce the release of NETs. Human neutrophils (Nϕ) were incubated with (**A**) amastigotes at the different MOIs for 90 min and NET-DNA quantified in the culture supernatant. Data were normalized according to spontaneous DNA release (Ctrl) and are shown as mean ± SEM of four independent experiments; *n* = 10; * *p* < 0.01, **** *p* < 0.0001. Black circles represent individual donors. (**B**) Representative image showing NETs and amastigotes stained with panoptic kit. Arrows point to NET-trapped amastigotes. Bar: 50 µm. (**C**–**F**) Neutrophils were incubated with CSFE-labeled amastigotes (green) for 90 min. Cultures were fixed and stained for elastase ((**E**); red). (**F**) Merged image of (**C**–**E**). Images representative of two experiments. Bars: 20 µm.

**Figure 3 tropicalmed-08-00336-f003:**
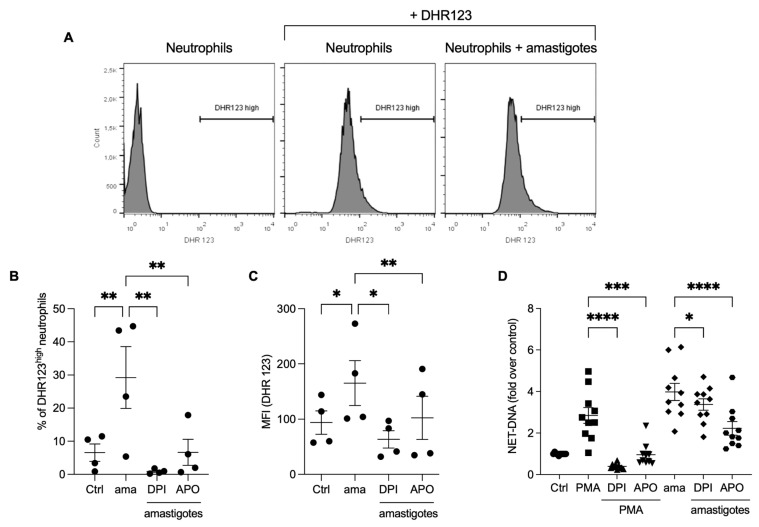
Amastigotes induce reactive oxygen species (ROS) production in neutrophils, which contributes to the NET release. (**A**–**C**) Human neutrophils were pretreated in the absence or presence of DPI or apocynin (APO) for 30 min, and then incubated with amastigotes or PMA for 20 min. (**A**) Flow cytometry gating strategy showing the neutrophil population we considered as high producers of ROS (DHR123^high^). Neutrophils were identified by size (FSC) and granularity (SSC). ROS production was evaluated using flow cytometry with DHR 123 probe. (**B**) Results expressed as the percentage of DHR123 high neutrophils and (**C**) as the mean fluorescence intensity (MFI) of 100,000 events analyzed. (**D**) Human neutrophils treated as above were incubated with amastigotes (AMA) or PMA for 90 min. NETs were quantified in the culture supernatant. Results normalized according to spontaneous DNA release (Ctrl) and presented as fold over control. Results shown as mean ± SEM of (**A**–**C**) two independent experiments (*n* = 4), and (**D**) six independent experiments (*n* = 10); * *p* < 0.05, ** *p* < 0.01, ****p* < 0.001 and **** *p* < 0.0001. (B-D) Symbols represent individual donors. (**D**) Circles (ctrl), squares (PMA), triangle (DPI + PMA), inverted triangles (APO + DPI), diamonds (ama and DPI + ama) and hexagons (APO + ama).

**Figure 4 tropicalmed-08-00336-f004:**
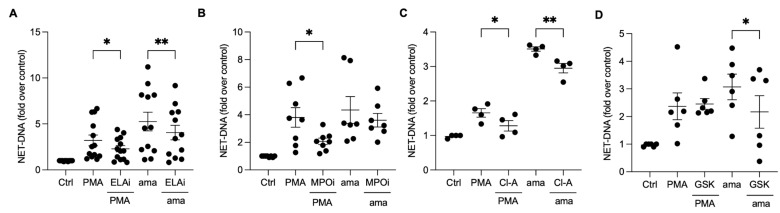
Neutrophil elastase and peptidyl arginine deiminases contribute to amastigote-induced NET release. Human neutrophils were pretreated or not (Ctrl) with (**A**) elastase inhibitor (ELAi), (**B**) myeloperoxidase inhibitor (MPOi), (**C**) chloroamidine (Cl-A), or (**D**) GSK-484 (GSK) for 30 min and then incubated with amastigotes (ama) or PMA for 90 min at 35 °C, 5% CO_2_. NETs were quantified in the culture supernatant using the PicoGreen kit. Data normalized according to spontaneous DNA release (Ctrl) are shown as mean ± SEM of (**A**) three independent experiments (*n* = 7), (**B**) five independent experiments (*n* = 10), (**C**) three independent experiments (*n* = 4), and (**D**) three independent experiments (*n* = 6); * *p* < 0.05 and ** *p* < 0.01. Black circles correspond to individual donors.

**Figure 5 tropicalmed-08-00336-f005:**
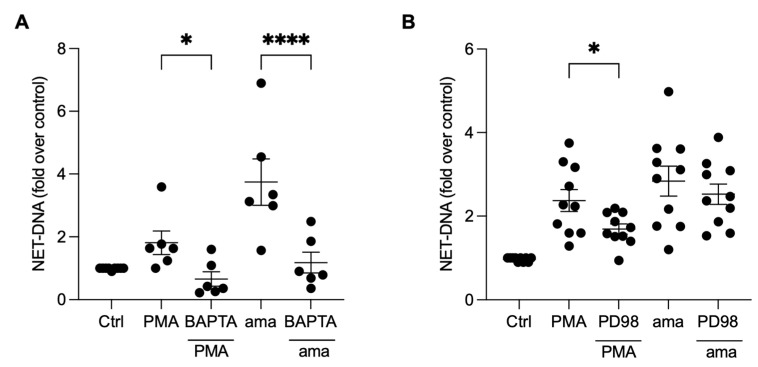
NET release induced by amastigotes requires calcium, but not ERK signaling. Human neutrophils were pretreated (**A**) in the absence or presence of BAPTA or (**B**) PD98 for 30 min and then incubated with amastigotes (ama) or PMA for 90 min at 35 °C and 5% CO_2_. NETs were quantified in the culture supernatant using the PicoGreen kit. Data normalized according to spontaneous DNA release (Ctrl) are shown as mean ± SEM of (**A**) three independent experiments (*n* = 6) and (**B**) four independent experiments (*n* = 10); * *p* < 0.05 and **** *p* < 0.0001. Black circles represent individual donors.

**Figure 6 tropicalmed-08-00336-f006:**
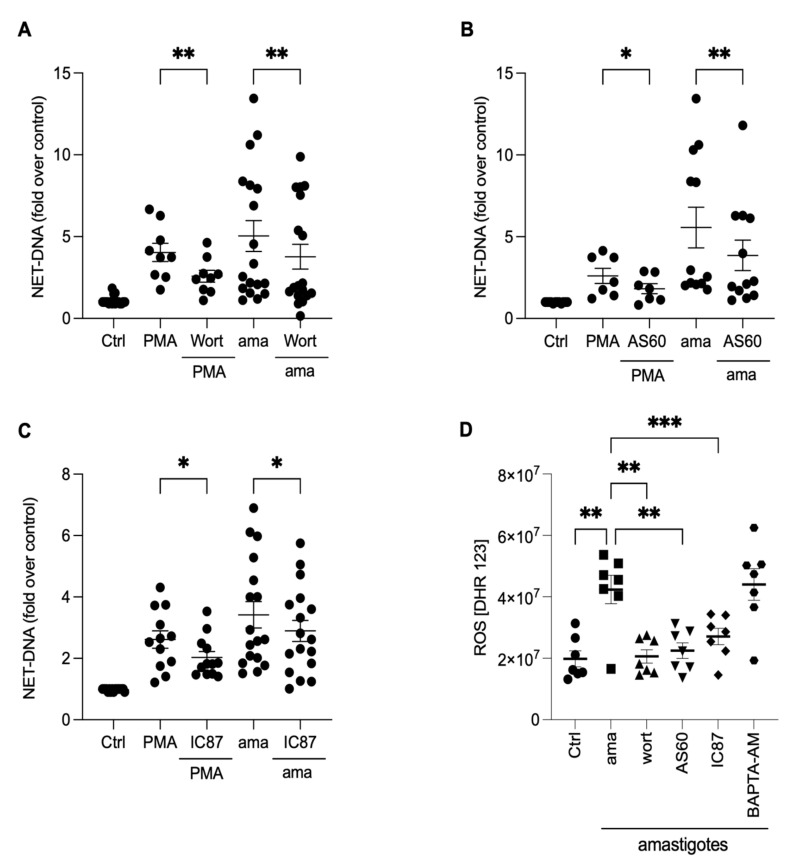
NETs induced by amastigotes depend on total PI3K, PI3Kγ, and PI3Kδ signaling pathways. Human neutrophils were pretreated in the absence or presence of (**A**) wortmannin (Wort), (**B**) AS60 (PI3Kγ inhibitor), or (**C**) IC87 (PI3Kδ inhibitor) for 30 min and then incubated with amastigotes (ama) or PMA for 90 min at 35 °C, 5% CO_2_. NETs were quantified in the culture supernatant. Data normalized according to spontaneous DNA release (Ctrl) and shown as mean ± SEM of (**A**) eight independent experiments (*n* = 18) and (**B**) seven independent experiments (*n* = 15); * *p* < 0.05, ** *p* < 0.01. (**D**) After treatment with inhibitors, cells were incubated with amastigotes for 120 min. ROS production was evaluated with the DHR 123 probe and analyzed using a spectrofluorometer reader. Results expressed as arbitrary units. Results shown as mean ± SEM of three independent experiments (*n* = 7); * *p* < 0.05, ** *p* < 0.01, and *** *p* < 0.001. (**A**–**D**) Symbols represent individual donors. (**D**) Circles (ctrl), squares (ama), triangle (wort + ama), inverted triangles (AS60 + ama), diamonds (IC87 + ama) and hexagons (BAPTA + ama).

**Figure 7 tropicalmed-08-00336-f007:**
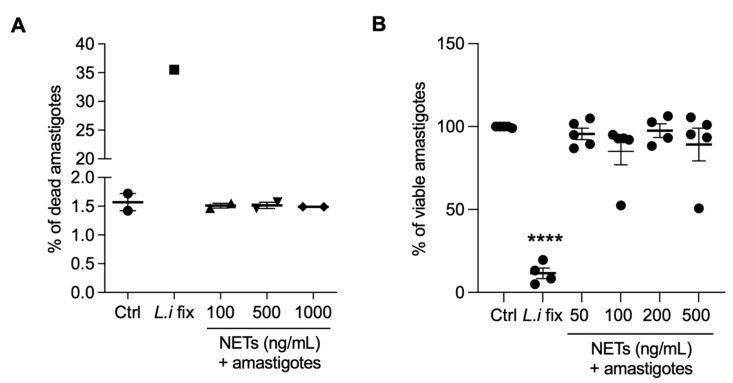
NETs are not toxic to axenic amastigotes. Amastigotes were incubated with NETs at the indicated concentrations, at 35 °C, 5% CO_2,_ for 5 h and analyzed using (**A**) flow cytometry or (**B**) alamar blue assay. Results expressed as the percentage of (**A**) dead amastigotes out of 100,000 analyzed events and as (**B**) viable amastigotes. Data shown as mean ± SEM of (**A**) two independent experiments and (**B**) five independent experiments; **** *p* < 0.0001. L.i fix: Leishmania infantum fixed in 4% formaldehyde. (**A**,**B**) Symbols correspond to different experiments.

## Data Availability

The data supporting the findings of this study are available within the paper or from the corresponding authors upon request.

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
