# Peer review of "Leishmania infantum Axenic Amastigotes Induce Human Neutrophil Extracellular Traps and Resist NET-Mediated Killing"

_tropicalmed, 2023, doi:10.3390/tropicalmed8070336_

Round 1
Reviewer 1 Report
1- The manuscript by Guimaraes-Costa et al. is well-written and conducted. However, before it is further processed, please clarify the real contribution of the obtained results to the development of these findings to improve how amastigotes activate innate immune responses and how help in the development of new and effective approaches to managing L. infantum infections. In addition, the authors must scientifically document the pathogenicity of L. infantum in any infectious age.
2- General, Reformulating the abstract in an orderly manner, highlighting the background of the objective, materials, method of work, results, and then the conclusion without writing it as a text according to the journal’s system
3- The introduction is adequate and comprehensive.
4- The method of work and materials used have been well documented.
5- Some results can be verified and discussed extensively.
6- Write conclusions, as they are not present at the end of the article.
Author Response
Point 1: The manuscript by Guimaraes-Costa et al. is well-written and conducted. However, before it is further processed, please clarify the real contribution of the obtained results to the development of these findings to improve how amastigotes activate innate immune responses and how help in the development of new and effective approaches to managing L. infantum infections. In addition, the authors must scientifically document the pathogenicity of L. infantum in any infectious age.
Response 1: We sincerely appreciate the reviewer's positive feedback on our paper. Our study primarily focuses on the analysis of amastigote activation of neutrophils, as it pertains to our basic science investigation. Given the complexity of the innate immune response involving numerous other factors, it becomes challenging to engage to speculate. Hence, we have modified the last sentence of the abstract to avoid such conjecture. Nevertheless, it is worth noting that existing literature indicates the immunomodulatory effects of neutrophil extracellular traps (NETs), which can potentially influence the activity of various innate immune cells. By stimulating the release of NETs, amastigotes may consequently modulate the behavior of other innate immune cells. We have previously demonstrated in our research that NETs can bias the differentiation of monocytes from dendritic cells towards anti-inflammatory macrophages, thereby increasing parasite survival (see [doi: 10.3389/fimmu.2017.00523]). Moreover, our group has provided evidence for the presence of NETs and trapped amastigotes within these structures in cutaneous leishmaniasis lesions (see [doi: 10.1371/journal.pone.0133063]). Interestingly, we observed the occurrence of extensive NET formations covering significant areas in certain lesions, which differed from those observed in other patients' lesions where the structures were thinner and smaller. In our study, we demonstrated a positive correlation between the number of amastigotes and the quantity of NETs present. Regrettably, we were unable to establish a direct correlation between these findings and disease severity or treatment refractoriness.
Several clinical trials are underway to control NET release or NET-mediated toxicity in diseases like COVID-19, sepsis, and ARDS. [e.g., Dornase Alfa Inhalation Solution [Pulmozyme] – phase 2; Safety and feasibility of inhaled rhDNase1 in severely ill COVID-19 patients- phase 1; Intravenous DNase I – sepsis – phase 1; Brensocatib, by blocking damaging neutrophil proteases – phase 3; Evaluate the effect of danirixin in reducing neutrophil extracellular traps (NETs) formation – phase 2]. Thus, it is possible that we could have strategies to control NET release/damage in the near future. Likewise, knowing the essential signaling pathways for NET release can help identify targets for interventions. Our challenge is determining whether eliminating NETs will be a good strategy for controlling cutaneous leishmaniasis.
In the present study we did not evaluate parasite infectivity. However, we possess data indicating that the axenic amastigotes of L. infantum employed in our research have demonstrated successful infection of murine and human monocyte-derived macrophages in vitro. Furthermore, the promastigotes utilized for the differentiation into amastigotes were obtained from infected mice to ensure their virulence. Moreover, our previous research has illustrated that dead Leishmania promastigotes can induce the release of NETs (see [doi: 10.3389/fimmu.2017.00523; doi: 10.1038/s41598-020-59486-2]). Consequently, the infectivity and viability of the parasites are dispensable for the induction of NET release. While it remains outside the scope of our current study, these previous findings suggest that even non-infective Leishmania parasites can elicit the release of NETs.
Point 2: General, Reformulating the abstract in an orderly manner, highlighting the background of the objective, materials, method of work, results, and then the conclusion without writing it as a text according to the journal’s system.
Response 2: We appreciate the reviewer's suggestion to reorganize the abstract in accordance with the journal's system [200 words only]. In our revised manuscript, we changed accordingly and the abstract now reads (the words in bold are just to highlight the added sections requested):
Introduction: Neutrophils are multifaceted cells that upon activation release meshes of chromatin associated with different proteins, known as neutrophil extracellular traps (NETs). Leishmania amazonensis promastigotes and amastigotes induce NET release, and we have identified the signaling pathways involved in NET extrusion activated by promastigotes. Amastigotes maintain the infection in vertebrate hosts, and we have shown the association of NETs with amastigotes in human biopsies of cutaneous leishmaniasis. However, the interaction of amastigotes and neutrophils remains poorly understood.
Objectives: Our study aimed to characterize the pathways involved in the formation of NETs induced by axenic amastigotes from L. infantum, the causal agent of visceral leishmaniasis.
Methodology: Human neutrophils pre-treated with signaling pathways inhibitors were incubated with amastigotes and NET release was quantified in the culture supernatant. Amastigotes viability was checked after incubation with NETs.
Results: We found that the release of NETs by neutrophils stimulated with these amastigotes requires the participation of elastase and peptidyl arginine deaminase, as well as the involvement of PI3K, ROS, and calcium. Moreover, amastigotes are not susceptible to NET-mediated killing.
Conclusion: Altogether, these findings improve our comprehension of the signaling pathways implicated in the interaction between amastigotes and human neutrophils.
Point 3: The introduction is adequate and comprehensive.
Point 4: The method of work and materials used have been well documented.
Point 5: Some results can be verified and discussed extensively.
Response 5: We appreciate the reviewer's request for a more detailed discussion of specific results. However, we kindly note that the reviewer did not explicitly specify which results they were referring to. Nevertheless, we respectfully disagree with the suggestion as we believe that our current discussion already provides comprehensive coverage of all the results obtained in our study. The discussion section of our paper aligns with the existing literature in the field and adequately addresses the findings in a thorough manner.
Point 6: Write conclusions, as they are not present at the end of the article.
Response 6: We appreciate the reviewer's observation regarding the absence of conclusions at the end of the article. As per the instructions provided by the journal, the inclusion of a conclusion section is optional. In our initial draft, we chose not to include a separate conclusion section based on these guidelines. However, we acknowledge the reviewer's concern and have taken their feedback into consideration. Consequently, in our revised manuscript, we have included a clear and concise conclusion section to summarize the main findings, addressing the reviewer's request. Thank you for highlighting this point, which prompted us to make the necessary adjustments.
Reviewer 2 Report
The manuscript is well-designed and interesting. However, I have the following minor comments as follows:
- Provide the primer accession number and product size.
- Many references are old and there are new versions of the manuscript. Please use the new papers instead of the old papers.
Author Response
Point 1: The manuscript is well-designed and interesting. However, I have the following minor comments as follows:
- Provide the primer accession number and product size.
Response 1: Thank you for your feedback. We apologize for the oversight in not including the primer accession numbers and product sizes. In our revised manuscript, we provided the primer accession numbers along with the corresponding product sizes for clarity and reproducibility of the experimental procedures. Please find the new information on page 3, Material and Methods section, at the end of "2.3. Axenic amastigote characterization" section.
Point 2: Many references are old and there are new versions of the manuscript. Please use the new papers instead of the old papers.
Response 2: We respectfully disagree with the reviewer's observation. In our manuscript, we have cited references based on their relevance and the critical knowledge they have contributed to the field, irrespective of their publication year. We have included seminal references, including the work of Brinkmann et al., 2004, which is widely recognized and foundational in the field. While we do not have specific clarity on the reference the reviewer is referring to, we have made adjustments in our revised manuscript by changing the other 2004 reference [#52] for a review published in 2023, in order to address the reviewer's concern. We believe that older references, particularly those that were the first to demonstrate the events mentioned in our text, still hold significant relevance in the field. Therefore, we have chosen to retain these references to acknowledge their foundational role in shaping the field of study.
Reviewer 3 Report
The manuscript “Leishmania infantum axenic amastigotes induce human neutro- 2 phil extracellular traps and resist NET-mediated killing” describes mechanisms underlying the release of NETs following activation with amastigotes of L. infantum. The authors demonstrated that this parasite induces NET release through PI3K activation, ROS production, calcium mobilisation, neutrophil elastase and PAD activation. They also showed that amastigotes are not susceptible to NET-mediated killing. The work is well written, the experiments well designed with clear and comprehensive graphs and tables. I think the target is suitable for the journal, and can be published in this form
Author Response
We sincerely appreciate the reviewer's positive feedback on our paper.